# Development and Validation of a Predictive Model Based on LASSO Regression: Predicting the Risk of Early Recurrence of Atrial Fibrillation after Radiofrequency Catheter Ablation

**DOI:** 10.3390/diagnostics13223403

**Published:** 2023-11-08

**Authors:** Mengdie Liu, Qianqian Li, Junbao Zhang, Yanjun Chen

**Affiliations:** 1Medicine School, Shenzhen University, Shenzhen 518000, China; 2110244091@email.szu.edu; 2Department of Cardiovascular Medicine, Peking University Shenzhen Hospital, Shenzhen 518000, China; 2210244077@email.szu.edu.cn (Q.L.); zjunb12134@foxmail.com (J.Z.)

**Keywords:** atrial fibrillation, radiofrequency ablation, early recurrence, LASSO regression, nomogram

## Abstract

Background: Although recurrence rates after radiofrequency catheter ablation (RFCA) in patients with atrial fibrillation (AF) remain high, there are a limited number of novel, high-quality mathematical predictive models that can be used to assess early recurrence after RFCA in patients with AF. Purpose: To identify the preoperative serum biomarkers and clinical characteristics associated with post-RFCA early recurrence of AF and develop a novel risk model based on least absolute shrinkage and selection operator (LASSO) regression to select important variables for predicting the risk of early recurrence of AF after RFCA. Methods: This study collected a dataset of 136 atrial fibrillation patients who underwent RFCA for the first time at Peking University Shenzhen Hospital from May 2016 to July 2022. The dataset included clinical characteristics, laboratory results, medication treatments, and other relevant parameters. LASSO regression was performed on 100 cycles of data. Variables present in at least one of the 100 cycles were selected to determine factors associated with the early recurrence of AF. Then, multivariable logistic regression analysis was applied to build a prediction model introducing the predictors selected from the LASSO regression analysis. A nomogram model for early post-RFCA recurrence in AF patients was developed based on visual analysis of the selected variables. Internal validation was conducted using the bootstrap method with 100 resamples. The model’s discriminatory ability was determined by calculating the area under the curve (AUC), and calibration analysis and decision curve analysis (DCA) were performed on the model. Results: In a 3-month follow-up of AF patients (*n* = 136) who underwent RFCA, there were 47 recurrences of and 89 non-recurrences of AF after RFCA. P, PLR, RDW, LDL, and CRI-II were associated with early recurrence of AF after RFCA in patients with AF (*p* < 0.05). We developed a predictive model using LASSO regression, incorporating four robust factors (PLR, RDW, LDL, CRI-II). The AUC of this prediction model was 0.7248 (95% CI 0.6342–0.8155), and the AUC of the internal validation using the bootstrap method was 0.8403 (95% CI 0.7684–0.9122). The model demonstrated a strong predictive capability, along with favorable calibration and clinical applicability. The Hosmer–Lemeshow test indicated that there was good consistency between the predicted and observed values. Additionally, DCA highlighted the model’s advantages in terms of its clinical application. Conclusions: We have developed and validated a risk prediction model for the early recurrence of AF after RFCA, demonstrating strong clinical applicability and diagnostic performance. This model plays a crucial role in guiding physicians in preoperative assessment and clinical decision-making. This novel approach also provides physicians with personalized management recommendations.

## 1. Introduction

Atrial fibrillation (AF) has become a prevalent sustained cardiac arrhythmia in clinical practice. Epidemiological studies have revealed that the incidence of AF affects between 2.3% and 3.4% of the population [1]. With an increasing aging population and the improvement of medical technology, the incidence and prevalence rate of AF are significantly increasing [2]. The effect of hemodynamic changes, progressing atrial and ventricular mechanical dysfunction, and thromboembolism contribute to an increased risk of stroke, heart failure, and increased mortality in patients with AF [3]. Radiofrequency ablation (RFCA) is currently the primary treatment for the recovery of sinus rhythm and prevention of complications in patients with AF that is not controlled by pharmacologic therapy and is difficult to treat [4,5,6].

However, the evidence suggests that patients experiencing AF recurrence after RFCA are not uncommon [7,8,9]. AF recurrence can be categorized into early recurrence (within 3 months after RFCA) and late recurrence (occurring more than 3 months after RFCA). Previous studies have found that the incidence of the early recurrence of AF (ERAF) is as high as 40–60% [10,11]. The 90 days after RFCA surgery are referred to as the “blank period”, during which, early recurrence of AF takes place. It has been proposed that early recurrence is associated with factors such as inflammatory damage during the ablation process [12], transient autonomic nervous system imbalances [13], and uneven scar tissue formation [14], but these associations lack specificity and are not often considered as actual clinical recurrences [15,16,17]. Nevertheless, many studies have demonstrated a connection between early and late recurrences after RFCA [18,19,20]. The concept of the “blank period” is still controversial, and early recurrence of AF should not be ignored [21,22]. In recent years, some studies have found early recurrence to be a strong predictor of late recurrence after radiofrequency ablation of atrial fibrillation [23]. The risk factors for early recurrence and the significance of early recurrence in AF are still unclear. Therefore, studying the risk factors for early recurrence in patients with AF after RFCA is of great significance for the long-term treatment of postoperative AF patients.

At present, most studies only identify factors that may affect atrial remodeling and involve the pathogenesis of AF as indicators for predicting the recurrence of AF. For instance, certain serum inflammatory markers such as C-reactive protein (CRP), interleukin-6 (IL-6), interleukin-2 (IL-2), endothelin-1, matrix metalloproteinase-2, and tumor necrosis factor-alpha have been considered relevant to the early recurrence of AF after RFCA [24,25,26,27]. However, the predictive value of these factors for AF recurrence remains controversial, and these biomarkers can only be reflective during the acute phase; as such, they cannot be used for the early identification of patients at high risk of AF recurrence after RFCA [28]. Currently, research on predicting the risk of postoperative recurrence in patients with AF through machine learning algorithms is very limited. Various risk scores for AF recurrence have been identified, but the discriminatory ability of these scores is highly variable and there are no widely used models to quantitatively predict AF recurrence in patients after RFCA [29]. In this study, we quantified preoperative parameters associated with early AF recurrence and employed the LASSO regression analysis to identify risk factors predicting early AF recurrence after RFCA. We developed and validated a novel predictive model to assess the risk of AF recurrence in patients undergoing RFCA, thereby equipping clinicians with additional potential tools to rapidly identify high-risk patients for early intervention and consequently reduce the recurrence rate of AF after RFCA.

## 2. Materials and Methods

### 2.1. Study Population

This study retrospectively analyzed the data of 150 patients with AF (including 95 with paroxysmal AF and 41 with persistent AF) who underwent their first RFCA procedure in the Department of Cardiology at Peking University Shenzhen Hospital from May 2016 to July 2022. All procedures were guided using the CARTO3 three-dimensional mapping system, and successful restoration of sinus rhythm was achieved. Inclusion criteria: Patients with a confirmed diagnosis of AF based on medical history, electrocardiography, or Holter monitoring (ESC 2020 AF guidelines) who underwent RFCA were included. Exclusion criteria: age < 18 or >90 years; presence of left atrial or left auricle thrombus detected by preoperative transesophageal echocardiography; severe valvular heart disease (severe mitral insufficiency and stenosis, severe tricuspid insufficiency and stenosis); severe heart failure (NYHA III or NYHA IV); malignant tumors in patients with a life expectancy of less than one year; hyperthyroidism; and chronic obstructive pulmonary disease. Among them, 136 individuals (90.7%) met the inclusion criteria and were included in this study (Figure 1).

This study has been approved by the Medical Ethics Committee (No. [2023] (078)) of Peking University Shenzhen Hospital, and all procedures were conducted in accordance with the principles of the Declaration of Helsinki and its subsequent amendments or comparable ethical standards.

### 2.2. Clinical Data

Basic baseline data of patients were collected, including age; gender; and whether they had hypertension, diabetes, coronary atherosclerotic heart disease, severe kidney disease (chronic renal failure IV, V), chronic obstructive heart disease, or severe cardiac insufficiency (cardiac function III, IV). Preoperative echocardiography (esophageal echocardiography), 12-lead electrocardiogram, 24 h Holter monitoring, and blood biochemical measurements were collected from patients.

### 2.3. Radiofrequency Catheter Ablation

All patients underwent RFCA under the guidance of the CARTO3 three-dimensional mapping system. A circular electrode was used to construct a complete left atrial anatomical model. Cold saline-solution-infused temperature-controlled ablation catheters were employed to perform circumferential isolation procedures on the left and right pulmonary veins. Ablation power was set to between 35 and 50 W, temperature was 43 °C, and the saline infusion rate was 17 mL/min. The ablation endpoint was achieved with electrical isolation of the pulmonary veins in the left atrium.

### 2.4. Postoperative Follow-Up and Recurrence

All patients underwent regular follow-up visits at the arrhythmia clinic at 1 week, 1 month, and 3 months after RFCA, or when symptoms of discomfort were found. During these visits, physical examinations and 12-lead electrocardiograms were performed. Patients experiencing symptoms of arrhythmia recurrence (such as palpitations, dizziness, etc.) were subjected to 24 h or 72 h Holter monitoring. If some patients were followed up at external institutions due to individual circumstances, telephone follow-up was conducted. The study endpoint was defined as the first documented occurrence of atrial tachyarrhythmia (atrial flutter, AF, atrial tachycardia) lasting >30 s within the initial three months following RFCA. Based on the 3-month postoperative follow-up records, patients were categorized into recurrence and non-recurrence groups.

### 2.5. Post-Analysis Variable Definitions

Based on the values of neutrophils (N), lymphocytes (L), monocytes (M), platelets (P), hemoglobin (Hb), and red cell distribution width (RDW) of preoperative patients, new inflammatory indices were determined as follows: the ratio of hemoglobin to red cell distribution width (HRR) was defined as Hb/RDW, the systemic immune inflammation index (SII) was defined as P × N/L, the neutrophil to lymphocyte ratio (NLR) was defined as N/L, the platelet to Lymphocyte ratio (PLR) was defined as P/L, and the lymphocyte to monocyte ratio (LMR) was defined as L/M.

### 2.6. Statistical Methods

Continuous variables were presented as the mean standard deviation or median and interquartile range, and group comparisons were performed using either the Student’s *t*-test or the Mann–Whitney *U* test. Categorical variables were expressed as frequencies and percentages, and group comparisons were conducted using the chi-squared test or Fisher’s exact test. Regarding the development of the predictive model, modern statistical shrinkage techniques, specifically, the least absolute shrinkage and selection operator (LASSO) regression, were employed to select factors related to the early recurrence of AF. The LASSO regression analysis is a shrinkage and variable selection method for linear regression models. In order to obtain the subset of predictors, the LASSO regression analysis minimizes the prediction error for a quantitative response variable by imposing a constraint on the model parameters that cause regression coefficients for some variables to shrink toward zero. Variables with a regression coefficient equal to zero after the shrinkage process are excluded from the model while variables with non-zero regression coefficient are most strongly associated with the response variable. Based on the type measure of −2log-likelihood and binomial family, the LASSO regression analysis running in R software runs the K cross-validation for the centralization and normalization of included variables 10 times and then picks the best lambda value. “Lambda.lse” gives a model with good performance but the least number of independent variables. So, the LASSO method was used to analyze the data in the training set to select the optimal predictors in the present risk factors. Then, multivariable logistic regression analysis was used to build a prediction model by introducing the feature selected in the LASSO regression model [30]. All of the selected features had statistical significance and were applied to develop the nomogram prediction models for he early recurrence of AF after RFCA. The discriminative ability of the model was determined by calculating the AUC. Internal validation was performed using bootstrap resampling (repeated 100 times). The model’s calibration was assessed using the Hosmer–Lemeshow test, and the clinical utility of the prognostic model was evaluated through DCA [31]. Statistical analyses were conducted using R software (version 4.3.1; R Foundation for Statistical Computing, Vienna, Austria). A significance level of *p* < 0.05 was considered statistically significant.

## 3. Results

### 3.1. Characteristics and Univariate Analysis of Recurrent AF after RFCA

After a 3-month follow-up, 47 patients experienced AF recurrence after RFCA, while 89 patients did not. Table 1 summarizes the demographic and clinical characteristics of the two groups. In the univariate analysis, the recurrence group exhibited significantly higher levels of P, PLR, RDW, LDL, and CRI-II compared to the non-recurrence group (*p* < 0.05). There were no statistically significant differences in baseline characteristics between the two groups for the remaining variables (*p* > 0.05).

### 3.2. Variable Selection Based on the LASSO Regression

The LASSO regression was used to establish the mathematical prediction model. According to the non-zero coefficients calculated by the LASSO regression analysis, the following four most powerful factors were determined: PLR, RDW, LDL, and CRI-II (Figure 2). Among these factors, RDW is the greatest risk factor for early recurrence of AF after RFCA, and the absolute value of this coefficient was the largest. The regression coefficients of each factor are shown in Table 2.

### 3.3. Development of the Model for Predicting the Risk of Early Recurrence of AF after RFCA

The results of the logistic regression analysis among PLR, RDW, LDL, and CRI-II are given in Table 3. All of these four predictors showed significant statistical differences. So, introducing the above four independent predictors, we constructed a nomogram based on PLR, RDW, LDL, and CRI-II to predict the risk of early recurrence of AF after RFCA. As is shown in the nomogram (Figure 3), with the increase in RDW, PLR, LDL, and CRI-II, the line chart score gradually increased, and the risk increased accordingly, indicating that the risk of early recurrence of AF increased after RFCA. In the nomogram model, the scores of RDW, PLR, LDL, and CRI-II were 100, 73, 60, and 35, respectively.

### 3.4. Prediction Model Validation

Subsequently, we generated the ROC curve for the predictive model. The AUC was 0.7248 (95% CI 0.6342–0.8155), highlighting the favorable diagnostic performance of the model (Figure 4A). Internal validation used the bootstrap method (resampling = 100), which resulted in an AUC of 0.8403 (95% CI 0.7684–0.9122) (Figure 4B). Moreover, the model exhibited good calibration, as indicated by a nonsignificant Hosmer–Lemeshow test result with a *p*-value of 0.239. This suggested that there was no statistically significant lack of fit between the predicted and observed values. To assess its clinical utility, we performed DCA (Figure 5). The DCA plot reveals that across the threshold probability range of 0.18–0.80, relative to “all individuals with AF recurrence” or “all individuals without AF recurrence”, the model provides a substantial net benefit.

## 4. Discussion

This study revealed an early recurrence rate of 34.6% (47/136) among AF patients after RFCA. We screened the risk factors for the early recurrence of AF by LASSO regression and developed a novel model to predict the risk of early recurrence of AF after RFCA. Multivariable logistic regression analysis was used to build a prediction model by introducing the feature selected in the LASSO regression model. In addition, a nomogram for predicting the recurrence of AF was established according to the standard procedure. This model has good discrimination and calibration. A nomogram is considered to be a reliable and practical predictive tool that can generate the individual probability of clinical events by integrating a different prognosis and different determinants [32], and quantify individual risk by combining a variety of important prognostic factors [33]. Notably, our research is the first to incorporate HRR, SII, CRI-I, and CRI-II into the predictive model for the early recurrence of AF after RFCA.

RFCA has become an important therapeutic strategy for improving symptoms and controlling heart rhythm [7,8]. However, the high recurrence rate of AF remains a significant challenge for clinicians. Some studies have found early recurrence to be a strong predictor of late recurrence after radiofrequency ablation of atrial fibrillation [23]. Therefore, we evaluated the individual risk of the early recurrence of AF before RFCA and identified patients at high risk of recurrence early, so as to guide clinical decision-making, minimize the postoperative recurrence rate of patients with AF, and improve the quality of life of patients.

Studies have shown that the level of inflammation is closely related to the occurrence and recurrence of AF [34,35]. Andrea Frustaci [36] found inflammatory markers in the atrial tissues of AF patients, including fibrosis, leukocyte infiltration, and oxidative damage. These inflammatory substances lead to electrical remodeling and structural reshaping of atrial tissues, potentially promoting the recurrence of AF [24,37]. Currently, most studies only look for factors that may influence atrial remodeling and are involved in the pathogenesis of AF, serving as indicators for predicting AF recurrence, for instance, some serum inflammatory biomarkers like CRP, IL-6, and IL-2. However, the predictive value of these factors for AF recurrence remains controversial, and these biomarkers can only be reflected in the acute phase, and they fail to early identify high-risk patients for AF recurrence. In this study, through an extensive literature review, we discovered several novel inflammation markers, including HRR, SII, CRI-I, and CRI-II, which can serve as significant predictive factors for cardiovascular diseases. We incorporated these variables into our study, employing the LASSO regression algorithm for variable selection and model establishment. Additionally, we developed a nomogram model for predicting AF recurrence after RFCA.

RDW is a blood parameter describing the change in red blood cell volume, which is mainly used in the differential diagnosis of anemia [38]. In recent years, RDW has been considered as a new inflammation predictor of cardiovascular disease [39,40]. Even in patients with lower CRP levels, increased RDW and cardiovascular disease mortality were statistically significant [41,42]. Increased RDW represents the biological processes of inflammation, aging, oxidative stress, nutritional deficiency, and impaired renal function [43]. It has been reported that an increase in RDW increases all-cause mortality and the occurrence of major adverse events in patients with AF [39,44,45]. In a prospective study, Kadri Murat Gurses et al. found that increased RDW may be a predictor of recurrence after cryoablation in AF patients [46]. To sum up, RDW may be an important predictor of the early recurrence of AF after RFCA.

The Castelli risk index (CRI) is a non-traditional lipid parameter that is calculated based on various lipoproteins in the blood. It is mainly used to evaluate the level of blood lipids clinically, and it has been proven to be associated with the inflammatory response in vivo [47]. It has been reported that lipid levels are involved in the occurrence and development of AF, but its pathophysiological mechanisms remain unclear. In vitro experiments have indicated that the lipid composition of the myocardial cell membrane surface might be involved in regulating the function of ion channels that initiate AF [48,49]. Traditionally, it has been believed that high levels of LDL cholesterol and TC, along with low levels of HDL cholesterol, are associated with the occurrence of AF [50]. However, in recent years, many studies have overturned this viewpoint. It was found that TC and LDL cholesterol are negatively correlated with AF [51]. The relationship between traditional lipid levels and AF still remains controversial. Compared with simple traditional lipid levels, CRI-I and CRI-II may be better at predicting the occurrence of cardiovascular adverse events.

SII is composed of P, N, and L, and is considered by researchers to be a new predictive index of inflammation that not only reflects the overall inflammatory state of the body, but also reflects the compensatory function of immune cells and the role of promoting coagulation [52]. It has been found that the increase in NLR, PLR, and LMR is associated with a new onset of AF and recurrence after RFCA [53,54,55], which can also lead to the development of atrial remodeling. Previous research has indicated that the prognostic significance of SII is more clinically valuable compared to its components (NLR or PLR) [56,57,58]. It is commonly used to predict adverse outcomes in various cancer patients [59]. In the past two years, SII has been shown to be associated with cardiovascular disease [60,61,62]. Studies have suggested that SII can serve as a predictive marker for AF occurrence following coronary artery bypass graft surgery [63]. However, there have been no reports on the correlation between SII and AF recurrence after RFCA. This study aims to explore the relationship between SII and early AF recurrence following RFCA.

In this study, we included N, L, M, P, Hb, NLR, PLR, LMR, TG, TC, HDL, and LDL as routine blood parameters as these parameters have been used to study the recurrence of AF. We also added some new inflammatory indicators, including RDW, HRR, CRI-I, CRI-II, and SII, which are considered to be closely related to the occurrence and development of cardiovascular disease. Although HRR, CRI-I, and SII failed to pass the screening of the LASSO regression, this study is the first to explore their relationship with the postoperative recurrence of AF. Multivariate logistic regression analysis of these four predictors found that they all showed statistically significant differences. We finally included four predictive factors: PLR, RDW, LDL, and CRI-II to create a nomogram predictive model. The model was internally validated (bootstrap method), demonstrating good discrimination and applicability, which makes our risk prediction more attractive in clinical practice. In our new model, all risk factors were obtained before RFCA, which means that the risk of the early postoperative recurrence of AF can be evaluated and predicted before RFCA, and further preventive measures can be taken before surgery to reduce the recurrence of AF and improve the long-term prognosis of patients.

## 5. Limitations

This study involved single-center research with a relatively small sample size, which may affect the generalizability of the findings. A multi-center study with a larger sample size should be conducted to further enhance the robustness of the results. Continuous rhythm monitoring was not performed after RFCA in patients, potentially missing asymptomatic recurrences. At present, there is evidence that the predictors of recurrent AF in the first 3 months are related to late recurrence [23]. Therefore, it is undoubtedly necessary to study the predictive factors of early AF recurrence in the future. Furthermore, external validation of the predictive model is required. Since there is no available 1-year follow-up data, it is not possible to compare long-term ablation success rates associated with recurrence during the blank period.

## 6. Conclusions

In summary, we have developed a novel preoperative nomogram prediction model based on the LASSO regression algorithm. This model incorporates four predictive factors and is designed to assess the risk of early postoperative recurrence in AF patients after RFCA. The model demonstrates strong discrimination and calibration capabilities, as well as a certain level of clinical applicability.

## Figures and Tables

**Figure 1 diagnostics-13-03403-f001:**
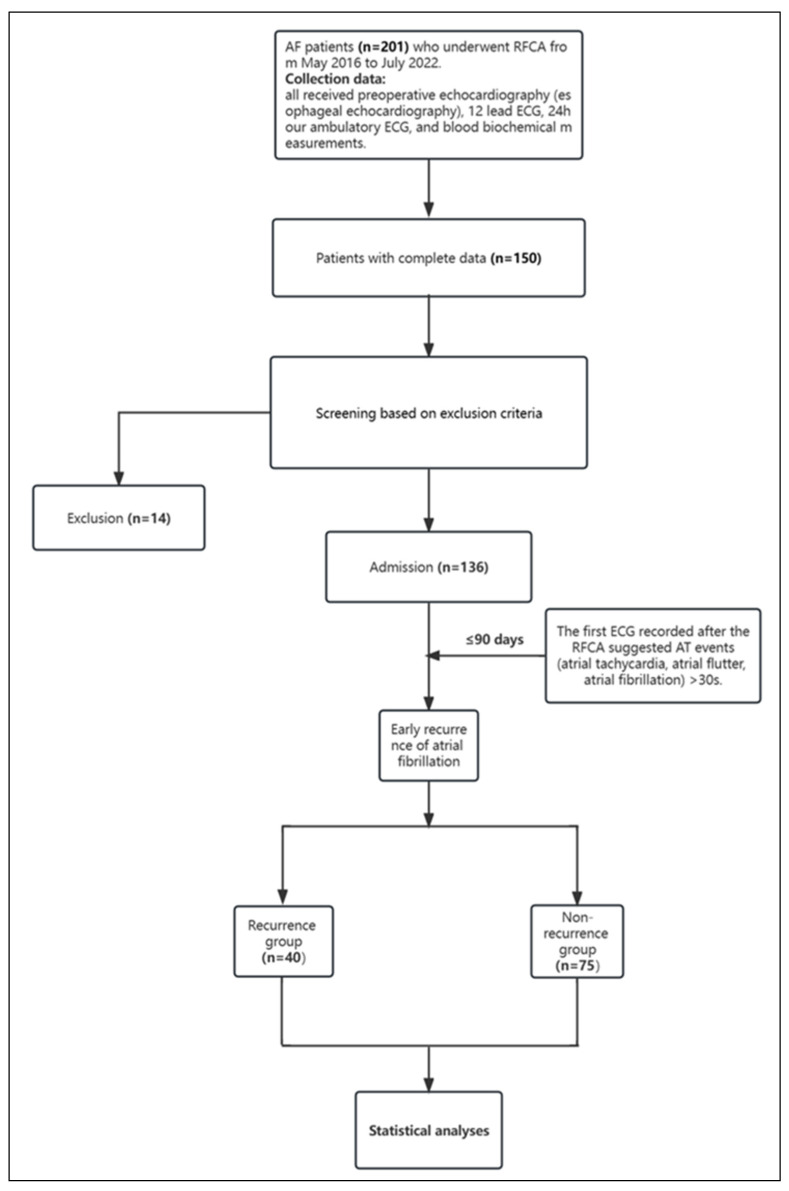
Flow of the Study.

**Figure 2 diagnostics-13-03403-f002:**
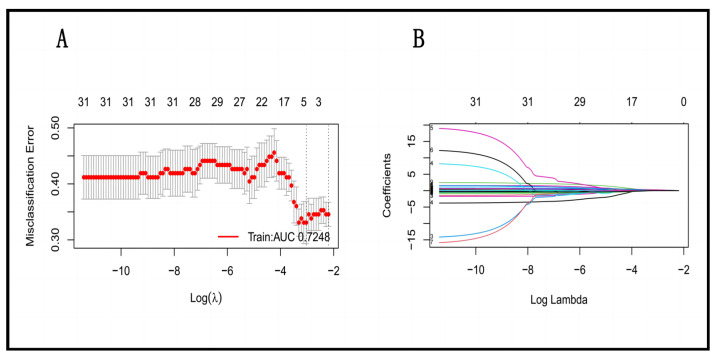
Results of a one-time randomized screening by LASSO regression. (**A**) Variation in misclassification error. The horizontal axis shows log λ and the vertical axis shows the misclassification error. The numbers above the curve represent the number of features with non-zero coefficient. The left dotted line represents the feature number corresponding to 0 standard error of misclassification. (**B**) The shrinkage plot of coefficients. The horizontal axis shows log λ and the vertical axis shows coefficient. The numbers above the curve represent number of features with non-zero coefficient.

**Figure 3 diagnostics-13-03403-f003:**
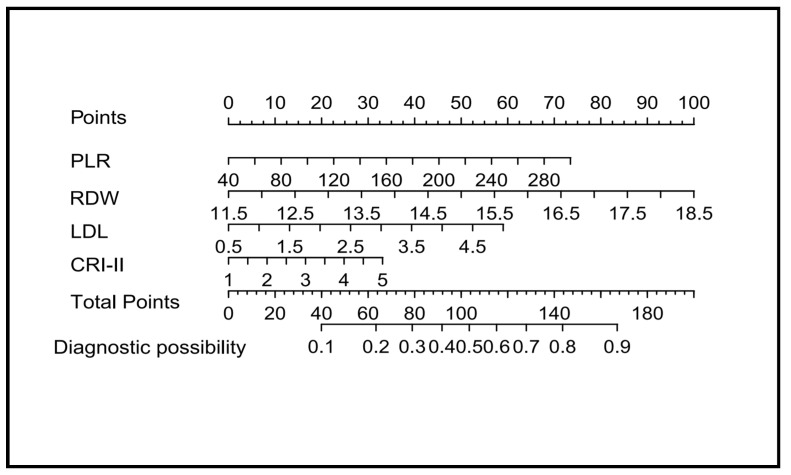
Nomogram for predicting early recurrence of AF at 3 months after RFCA.

**Figure 4 diagnostics-13-03403-f004:**
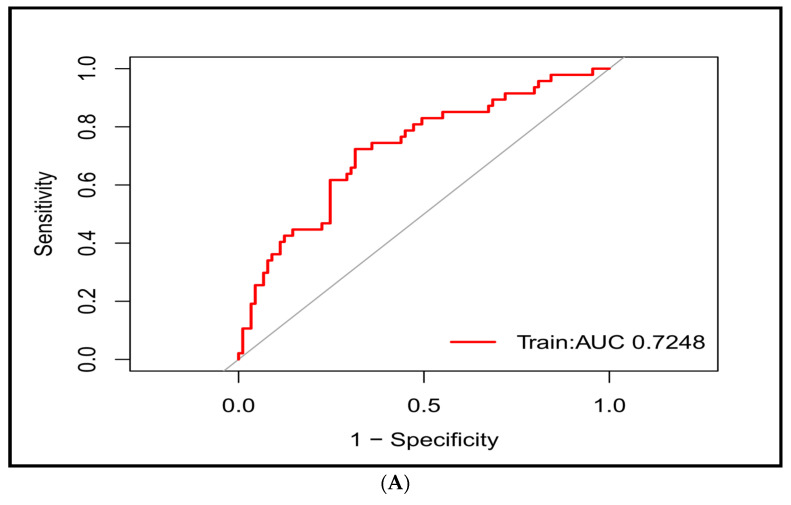
(**A**) The ROC curve of the prediction model (representing the discrimination ability of the model). AUC is 0.7248 (95% CI 0.6342–0.8155). (**B**) The ROC curve for bootstrap internal verification (resampling = 100). AUC is 0.8403 (95% CI 0.7684–0.9122).

**Figure 5 diagnostics-13-03403-f005:**
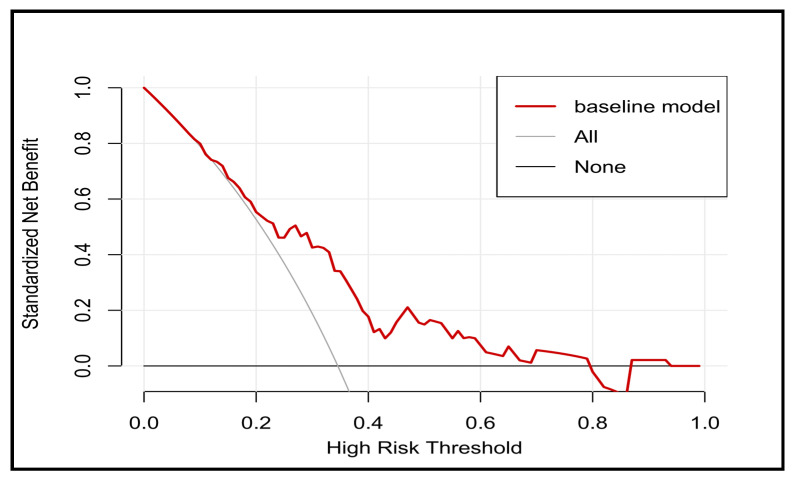
The DCA of the nomogram. The DCA shows that when the threshold probability is between 0.18 and 0.8, compared with “all individuals with recurrent AF” or “individuals without recurrent AF”, the application of this line chart will increase the net benefit.

**Table 1 diagnostics-13-03403-t001:** Baseline and clinical characteristics (N = 136).

	Rhythm after Surgery (3 m)		*p*-Value
	Recurrence (n = 47)	Non-Recurrence (n = 89)	
Age (year)	61.09 ± 11.88	59.51 ± 11.41	0.456
Gender (n, %) (Male versus female)	31 (34.8) versus 16 (34.0)	58 (65.2) versus 31 (66.0)	1.000
Hypertension (n, %)	19 (40.4)	39 (43.8)	0.843
Coronary artery disease (n, %)	7 (14.9)	11 (12.4)	0.882
Diabetes mellitus (n, %)	8 (17.0)	14 (15.7)	1.000
Severe renal dysfunction (n, %)	3 (6.4)	4 (4.5)	0.947
Severe cardiac insufficiency (n, %)	13 (27.7)	22 (24.7)	0.868
COPD (n, %)	0 (0.0)	0 (0.0)	-
N	4.12 ± 1.75	3.90 ± 1.41	0.419
L	1.84 ± 0.53	1.93 ± 0.63	0.392
M	0.44 ± 0.15	0.46 ± 0.19	0.725
P	226.57 ± 65.65	202.85 ± 57.33	**0.031**
Hb	138.30 ± 18.20	137.90 ± 15.43	0.893
NLR	2.38 ± 1.28	2.22 ± 1.13	0.474
PLR	130.17 ± 42.92	111.64 ± 36.53	**0.009**
LMR	4.42 ± 1.55	4.75 ± 2.49	0.398
SII	540.64 ± 360.91	448.47 ± 261.72	0.090
RDW	13. 17± 1.48	12.62 ± 0.84	**0.006**
HRR	10.66 ± 1.93	11.00 ± 1.57	0.272
TG	1.28 ± 0.52)	1.39 ± 0.84	0.416
TC	4.36 ± 0.91	4.09 ± 1.04	0.136
HDL	1.11 ± 0.23	1.13 ± 0.28	0.816
LDL	2.95 ± 0.76	2.64 ± 0.79	**0.030**
CRI-I	4.00 ± 0.91	3.73 ± 0.94	0.117
CRI-II	2.71 ± 0.79	2.42 ± 0.78	**0.044**
Antiplatelet drugs	18 (20.2)	6 (12.8)	0.396
Βeta-blockers	37 (41.6)	6 (12.8)	0.522
ACEI/ARBs	39 (83.0)	69 (77.5)	0.600
Calcium channel blockers	7 (14.9)	4 (15.7)	1.000
Statin drugs	25 (53.2)	55 (61.8)	0.431

**Table 2 diagnostics-13-03403-t002:** LASSO regression screening for AF recurrence outcome performed once at random.

Factors	LASSO Coefficient
PLR	0.002958548
RDW	0.148474080
LDL	0.127461973
CRI-II	0.002393739

**Table 3 diagnostics-13-03403-t003:** Predictors for the risk of early recurrence of AF after RFCA.

	OR	95% CI	*p*-Value
PLR	1.002	1.000–1.004	0.048
RDW	1.102	1.028–1.181	0.006
LDL	1.134	1.029–1.249	0.012
CRI-II	1.592	1.006–2.520	0.047

## Data Availability

The data presented in the study are available on reasonable request from the corresponding author. The data are not publicly available due to respect for patients’ privacy.

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
