# Peer review of "Development and Validation of a Predictive Model Based on LASSO Regression: Predicting the Risk of Early Recurrence of Atrial Fibrillation after Radiofrequency Catheter Ablation"

_diagnostics, 2023, doi:10.3390/diagnostics13223403_

Round 1

Reviewer 1 Report

Comments and Suggestions for Authors

Development and validation of a predictive model based on LASSO regression: Predicting the risk of early recurrence after radiofrequency catheter ablation

[1] Would it be better to use "early recurrence of Atrial Fibrillation" rather than just "early recurrence"? It seems strange and incomplete to just write "early recurrence" such as in the title and in the other parts of the manuscript. Early recurrence of what?

[2] What are the effects of age, gender, BMI, socio-economic status, etc., on the early recurrence of Atrial Fibrillation?

[3] Was the initial severity of the patient's condition considered in the regression model?

[4] The authors should justify the use of LASSO regression. Why was LASSO regression selected? Would other regression methods have worked just fine or even better? What do the results look like with other (regression) methods?

[5] Were the assumptions for the regression model tested?

[6] What variables were included in the model? Did the authors use only the variables in Table 2? How did the authors control the effects of other variables listed in Table 1, especially since some are potential confounders?

[7] Were the variables normalized before they were passed as input into the regression model? If not, would it be better to do so to better account for the relative importance of the variables (or why not)?

[8] The horizontal axis for the ROC curve in Fig 4B should be limited to the range 1.0 to 0.0 [1, 0.0].

Comments on the Quality of English Language

The English usage by itself is good. However, some important information are missing. Please refer to my numbered comments above.

Author Response

Dear Reviewer, The revision notes have been submitted in Word version.

Reviewer 2 Report

Comments and Suggestions for Authors

In the article authors developed a novel preoperative mathematical prediction model based on the LASSO regression. This developed model incorporates four predictive factors and is designed to assess with good results the risk of early postoperative recurrence in AF patients after RFCA.

The idea of article is quite interesting as the developed model demonstrated strong predictive capability suitable, along with favorable calibration, for clinical applications. Few minor concerns:

-          Although currently research on predicting the postoperative recurrence in patients with AF through ML models is limited, few similar approaches have to taken into account for a rigorous comparative analysis;

-          More details about the mathematical prediction model presented in the article are necessary. Now the developed model contains only one model equation used for the risk index of early postoperative recurrence for each patient;

-         Also more details are necessary for the nomogram predicting early recurrence of AF at 3 months after RFCA, presented in Fig 3., as it is represented the main key of the developed prediction model;

-          More patients with atrial fibrillation after radiofrequency catheter ablation are needed to be enrolled in the study to be statistically relevant.

Author Response

(The authors gave the same response as above.)

Reviewer 3 Report

Comments and Suggestions for Authors

General comments:

The purpose of the study is to create and validate a predictive regression model that can determine the risk of early recurrence of AF. The prediction model achieved an AUC of 0.72, which increased to 0.84 with Bootstrap. The model has significant potential for clinicians to identify high-risk patients for early recurrence of AF. However, the study acknowledged some critical limitations to the model. One potential limitation was the lack of continuous rhythm monitoring in the study design. Furthermore, the model should be compared with machine learning using the same dataset. To emphasize the significance of detecting early recurrence of AF, the study may need to add the relationship between early recurrence and late recurrence of AF to the Introduction section (second paragraph). For example, "Recent studies showed that early recurrence is a reliable predictor of late recurrence after RFCA of AF [PMID: 33516711]."

Specific comments:

1) Change "provide" to "provides" in line 37.

2) The sentence in lines 37 and 38 needs to be changed. The study is not a machine learning (ML) approach, but a LASSO regression.

3) Include "the early recurrence of AF" (ERAF) in line 55.

4) Please add a reference in line 75.

5) What is the patient number for paroxysmal and persistent AF? Is there any difference between the two groups for early recurrence of AF in Table 1?

6) The 95% CI number (0.729-0.798) of AUC in line 193 is different from the abstract: 0.7248 (95% CI 0.6342-0.8155) in line 30 and figure 4A. Which is the correct CI?

7) Remove "Chinese" from Figure 4B.

8) Change "baseline mudel" to "baseline model" in Figure 5.

9) Change "proved" to "proven" in line 254.

10) Ensure the same terminology x and y-axis in Figures 4A and 4B. Correct the x-axis scale in Figure 4B.

11) Change "mechanism" to "mechanisms" in line 256.

12) Change "cholestero" to "cholesterol" in lines 259 and 261.

13) Add a reference in line 303.

14) Move the limitations section before the conclusion section.

Comments on the Quality of English Language

Need to change some typos.

Author Response

(The authors gave the same response as above.)
